materials science/high-pressure physics

pyrochlore, order–disorder phase transition, high pressure

**Author for correspondence:**
Jingjing Niu
e-mail: jjniu@itpcas.ac.cn

This article has been edited by the Royal Society of Chemistry, including the commissioning, peer review process and editorial aspects up to the point of acceptance.

# Pressure-induced order–disorder transition in Gd$_{1.5}$Ce$_{0.5}$Ti$_2$O$_7$ pyrochlore

Jingjing Niu[1,2,3], Xiang Wu[4], Haibin Zhang[3] and Shan Qin[2]

[1]Key Laboratory of Continental Collision and Plateau Uplift, Institute of Tibetan Plateau Research, Chinese Academy of Sciences, Beijing 100101, People's Republic of China
[2]Key Laboratory of Orogenic Belts and Crustal Evolution, MOE, Peking University and School of Earth and Space Sciences, Peking University, Beijing 100871, People's Republic of China
[3]Innovation Research Team for Advanced Ceramics, Institute of Nuclear Physics and Chemistry, China Academy of Engineering Physics, Mianyang 621900, People's Republic of China
[4]State key laboratory of geological processes and mineral resources, China University of Geosciences (Wuhan), Wuhan 430074, People's Republic of China

JN, 0000-0003-2290-8477

An experimental study on ordered pyrochlore structured Gd$_{1.5}$Ce$_{0.5}$Ti$_2$O$_7$ ($Fd\bar{3}m$) was carried out up to 45 GPa by synchrotron radiation X-ray diffraction and Raman spectroscopy. Experimental results show that Gd$_{1.5}$Ce$_{0.5}$Ti$_2$O$_7$ transfers to a disordered cotunnite-like phase ($Pnma$ Z = 4) at approximately 42 GPa. Compared with the end member Gd$_2$Ti$_2$O$_7$, the substitution of Ce$^{3+}$ for Gd$^{3+}$ increases the transition pressure and the high-pressure stability of the pyrochlore phase. This pressure-induced structure transition is mainly controlled by cationic order–disorder modification, and the cationic radius ratio $r_A/r_B$ may also be effective for predicting the pyrochlore oxides' high-pressure stability. Two isostructural transitions are observed at 6.5 GPa and 13 GPa, and the unit-cell volume of Gd$_{1.5}$Ce$_{0.5}$Ti$_2$O$_7$ as a function of pressure demonstrates its compression behaviour is rather complex.

## 1. Introduction

Pyrochlore oxide, with an ideal chemical formula of A$_2$B$_2$O$_6$O′(or A$_2$B$_2$O$_7$), have attracted substantial attention due to its unique structural properties and its applications in fuel cells [1,2], spin liquid materials [3] and high-level waste disposal materials [4]. The pyrochlore structure (figure 1a) belongs to the $Fd\bar{3}m$ space group (Z = 8). It can be viewed as A and B cation ordered 2 × 2 × 2 superlattices of ideal fluorite structures ($Fm3m$) with 1/8 anion deficiency. The larger A cations occupied 16c (1/2 1/2 1/2) with

**Figure 1.** The crystal structures of $A_2B_2O_7$ oxide: (*a*) the pyrochlore structure and (*b*) the cotunnite-like structure. The purple balls are $A^{3+}$ cations and the blue octahedrons are [$BO_6$]. Small red balls are oxygen ions. In the cotunnite-like $A_2B_2O_7$ structure, the A and B cations are disordered and 1/8 of the $O^{2-}$ vacancies are randomly distributed.

eight coordinates located in a distorted cubic polyhedron. The six-coordinate B site located in 16*d* (0 0 0) is usually occupied by smaller cations centred in an oxygen octahedron. The oxygen $O^{2-}$ anions occupy the 48*f* ($x$ 3/8 3/8) site, the $O'^{2-}$ anions occupy the 8*a* (1/8 1/8 1/8) site, and the 8*b* site is systematically vacant. Empirically, the structural stability of $A_2B_2O_7$ pyrochlores at ambient conditions depends on the ratio of the cation radii, $r_A/r_B$ [5], only when $1.46 < r_A/r_B < 1.78$, $A_2B_2O_7$ oxides crystallize in the pyrochlore structure.

As an important thermodynamic parameter, pressure can strongly affect the structures and properties of materials. The safe immobilization of toxic high-level nuclear waste (HLW) requires the waste forms are isolated from biosphere over time scales much longer than the span of recorded human history, because of the radiotoxicity of long half-life isotopes ($^{239}$Pu, half-life 24 000a). Collapse, explosion and geological changes may exert high pressure on the forms of HLW which are enclosed in the disposal repository at a depth of approximately 1000 m underground. Besides, the large radius actinides substituted into the pyrochlore lattice will decrease the phase stability of the lattice under pressure (e.g. pv-ppv transition pressure of $NaMgF_3$ and $MgSiO_3$ [6]). Above all, the phase stability of the substituted pyrochlore under high pressure needs to be considered.

The high-pressure behaviour of pure member pyrochlore oxides has been extensively investigated. Theoretical simulations reveal that titanate pyrochlores (B = $Ti^{4+}$) and zirconated pyrochlore (B = $Zr^{4+}$) can transfer to an orthorhombic cotunnite-like structure (*Pnma* and Z = 4) (figure 1*b*) at high pressure [7–9]. At 11 GPa and 1300°C, pyrochlore $Eu_2Ti_2O_7$ transfers to a perovskite-like structure ($P2_1$, denoted PL-$Eu_2Ti_2O_7$), which has been confirmed as a high-temperature ferroelectric material [10]. In addition, the substitution of pyrochlore-type oxides can form complex composition pyrochlore oxides, which helps control their structure and physical properties. In the $Gd_2Ti_{2-x}Zr_xO_7$ binary system, the transition pressure increases along with the decrease of the B site substituting $Zr^{4+}$ [11], while the transition pressure slightly changes in zirconated pyrochlore ($Gd_{0.9}U_{0.1}$)($Zr_{0.9}U_{0.1}$)$O_{7+\delta}$ with U-doped in both the A site and B site [12]. $Ce^{3+}$ is often used in research as a nonradioactive surrogate for Pu because they share common chemical and crystal-chemical properties. Zhang *et al*. revealed that the $Gd_{2-x}Ce_xTi_2O_7$ system maintains pyrochlore structures when $x < 0.8$ [13]. Here, we chose $Ce^{3+}$ substitution for $Gd^{3+}$ to increase the A site average cationic radius, and carried out the experimental study on $Gd_{1.5}Ce_{0.5}Ti_2O_7$ up to approximately 40 GPa, in order to explore the $Ce^{3+}$-doping influence on its phase stability and high-pressure behaviour.

## 2. Experimental details

The sample in the present study was synthesized using a combustion method. The starting materials tetrabutyl titanate [$Ti(OBu)_4$] (Aladdin, greater than 99.0%), $Gd(NO_3)_3 \cdot 6H_2O$ (Aladdin, 99.9%) and $Ce(NO_3)_3 \cdot 6H_2O$ (Aladdin, 99.9%), were dissolved stoichiometrically in nitric acid and deionized water, respectively, with magnetic stirring. Glycine (Aladdin, 99%) as a fuel with a mole ratio $n(Gly)/n(Ti) = 2.8$ was added to the mixed solution. This mixture was heated on a hot plate until an auto-ignition process in a corundum crucible. The obtained solid was sintered at 1473 K for 2 h under Argon atmosphere in order to avoid the $Ce^{3+}$ being oxidized.

Symmetry-type diamond anvil cells were employed as a high-pressure apparatus. Two runs of *in situ* synchrotron X-ray diffraction experiments were carried out under $\lambda = 0.6199$ Å. Rhenium gaskets were

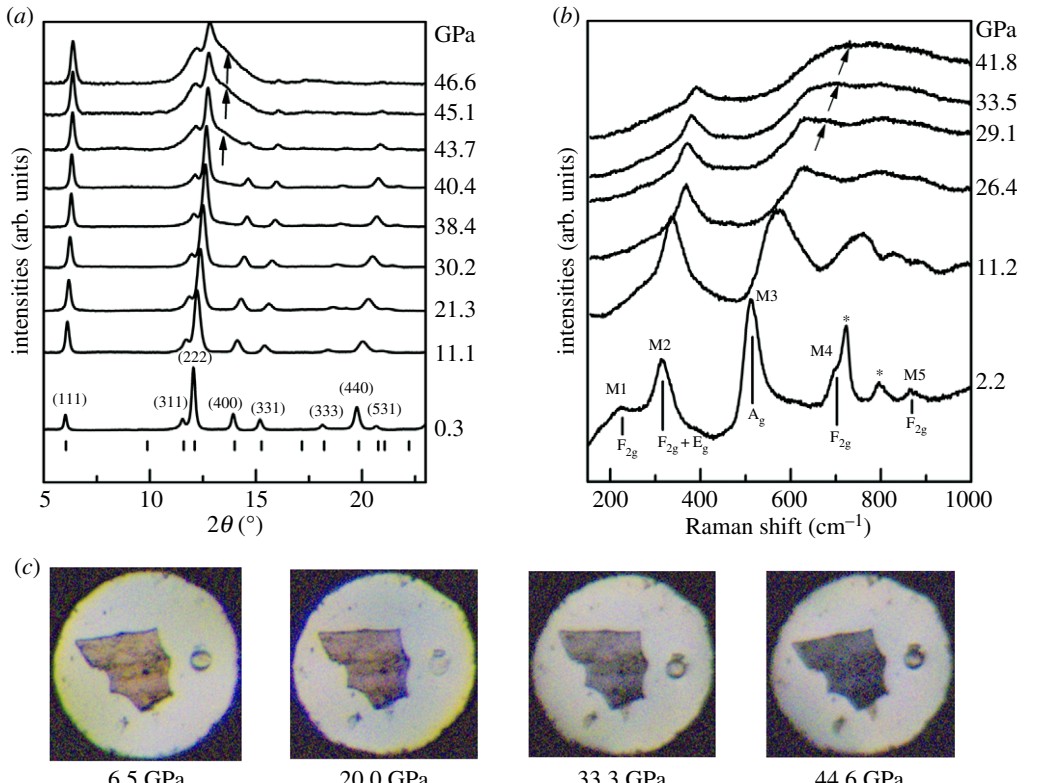

**Figure 2.** (*a*) The *in situ* synchrotron high-pressure XRD patterns of run 1. The arrow indicates that the new reflections belong to the *Pnma* cotunnite-like phase. (*b*) The *in situ* high-pressure Raman spectra of the $Gd_{1.5}Ce_{0.5}Ti_2O_7$. The star (*) mode belongs to the silicone oil, and the arrows indicate the new band observed at approximately 700 cm$^{-1}$ when the pressure is higher than 29.1 GPa. (*c*) The sample's colour change under different pressures.

pre-indented to approximately 40 μm in thickness with a hole approximately 150 μm in diameter in the centre of the indentation as sample chambers. Run 1 was performed at the Shanghai Synchrotron Radiation Facility (SSRF) BL15U1 beamline at a pressure up to approximately 47 GPa. The pressure transmitting media was silicone oil. A slice of Au (99.9%, Alfa Aesar, Haverhill, MA, USA) was loaded into the sample chamber, and its equation of state (EoS) was used to determine the pressure [14]. In run 2, an experiment up to approximately 20 GPa was carried out at the Beijing Synchrotron Radiation Facility (BSRF) 4W2 beamline. Noble gas argon was loaded into the sample chamber as PTM and the pressure was monitored using the ruby fluorescence method [15]. All of the XRD patterns were converted from Debye rings to one-dimensional X-ray profiles versus $2\theta$ via FIT2D code [16]. The high-pressure XRD patterns were fitted using the Le Bail method implemented using GSAS + EXPGUI software [17].

High-pressure Raman experiments were carried out up to approximately 40 GPa at room temperature on a Renishaw inVia reflex laser Raman spectrometer. A 532 nm diode-pumped solid-state laser was employed as the excitation light source. The polycrystalline sample was compressed into slices and placed in a 150 μm diameter hole drilled in pre-indented Rhenium gaskets. The ruby fluorescence technique was employed to calibrate the pressure [15], and silicone oil was used as the pressure medium.

# 3. Results

Figure 2*a* shows the selected XRD patterns from run 1 of the $Gd_{1.5}Ce_{0.5}Ti_2O_7$ under different pressure. The patterns of run 2 are available in the supplementary materials (electronic supplementary material, figure S1). At the beginning of the experiments, all of the reflections can be indexed as a pyrochlore structure, indicating that the $Gd_{1.5}Ce_{0.5}Ti_2O_7$ crystallized in the pyrochlore structure. At 43.7 GPa, a new reflection arises at approximately 13° between the (222) and (400) reflections of the pyrochlore structure. The intensities of (400), (331), (333), (440) and (531) reflections decrease while the intensity of (111) increases. The sample undergoes a pressure-induced phase transformation to an orthorhombic

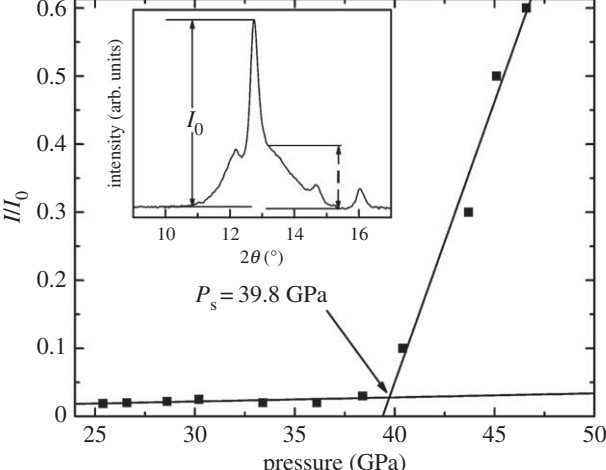

**Figure 3.** The intensity ratio between (222) of the pyrochlore and cotunnite phase as a function of pressure. Cotunnite intensity is defined as the intensity between the (222) and (004) pyrochlore reflections.

cotunnite-like phase (*Pnma*). Due to the high degree of disorder and large strain inherent in the high-pressure cotunnite-like phase, it is difficult to identify the structure of the high-pressure phase through XRD. So an absolute phase fraction of cotunnite-like phase in the sample is hard to refine by the Rietveld method. However, the intensity ratio ($I/I_0$) between the scattering intensity between the (222) and (004) pyrochlore structure diffraction ($I$), which is the location of the most intense cotunnite peaks and the intensity of (222) reflection ($I_0$), is employed as a relative phase fraction in order to determine the onset transition pressure. The $I/I_0$ as a function of pressure are shown in figure 3. After approximately 40 GPa, the $I/I_0$ increases rapidly, and the onset of the transition pressure is 39.8 GPa. Here, in order to compare with previous studies, the onset transition pressure is determined as 42(2) GPa based on the intensity increase of the cotunnite-like reflections.

According to group theory, pyrochlore structured $Gd_{1.5}Ce_{0.5}Ti_2O_7$ has 6 Raman active modes, namely,

$$\Gamma = A_{1g} + E_g + 4F_{2g}. \tag{3.1}$$

Raman spectra from $Gd_{1.5}Ce_{0.5}Ti_2O_7$ recorded from 150 cm$^{-1}$ to 1000 cm$^{-1}$ at various pressures are shown in figure 2*b*. At 2.2 GPa, five vibration modes can be identified and labelled as M1 to M5. According to previous studies [18–20], M1, M4 and M5 were assigned as $F_{2g}$ and M3 as $A_g$. M2 contains two modes that are assigned as $F_{2g} + E_g$ with close frequencies. The vibrations of the pressure transmitting medium (PTM) silicone oil are marked with stars (*). At 29.1 GPa, a new band appears at approximately 700 cm$^{-1}$, and it is believed to be related to the distortion of the [$TiO_6$] octahedron [21,22]. The [$TiO_6$] distortion also causes the colour of the sample to change under pressure (figure 2*c*). At approximately 30 GPa, the colour of $Gd_{1.5}Ce_{0.5}Ti_2O_7$ changes from transparent orange-ish to dark purple-blue. Combined with the XRD pattern and Raman spectra, this colour change may be due to the distortion of the [$TiO_6$] octahedron rather than the phase transition.

## 4. Discussion

Apart from crystal structure prediction under ambient conditions, the cationic radius ratio $r_A/r_B$ is effective for predicting the high-pressure stability of pyrochlore oxides. The transition pressure of $Gd_{1.5}Ce_{0.5}Ti_2O_7$ is similar to that of $Eu_2Ti_2O_7$ [23] and $Sm_2Ti_2O_7$ [24] but larger than that of $Gd_2Ti_2O_7$ [11]. The transition pressures of titanite pyrochlore oxides are listed in table 1. A larger ionic radius replacement will usually lower the transition pressure. However, the current study found that as the radius of the A-site cation increased, the pyrochlore to cotunnite-like phase transition pressure rose (figure 4). This unusual tendency is related to the mechanism of the pyrochlore-cotunnite transition. The high-pressure cotunnite-like phase is a highly disordered phase. First, unlike ideal cotunnite-structured oxides ($AO_2$), A and B cations are disordered in the cotunnite-like $A_2B_2O_7$, and 1/8 of $O^{2-}$ are randomly vacant. Second, the cotunnite-like high-pressure phase likely causes many disordered anion vacancy defects due to the vacancies present in the pyrochlore structure. From this

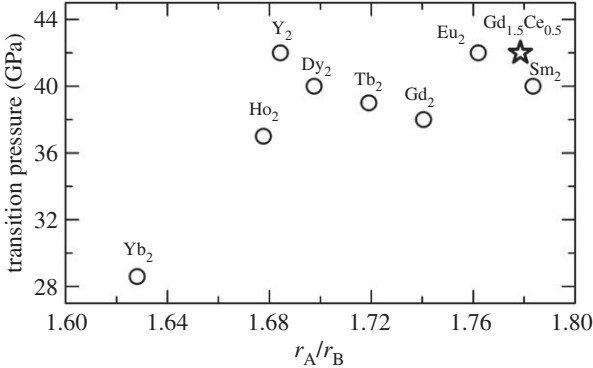

**Figure 4.** Correlation plot between the transition pressure to a cotunnite-like structure for titanite pyrochlores and the cationic radii ratio ($r_A/r_B$). The ionic radii of the A-site REE are the value of the eightfold coordinated cations with the chemical valence of +3, and $Ti^{4+}$ ionic radius is the value of the sixfold coordinated cations. The round black marks represent the transition pressure listed from previous research (table 1) and the star represents the results of this study.

**Table 1.** The titanite pyrochlore ($A_2Ti_2O_7$) to cotunnite-like transition pressure versus the cationic radius ratio. The cationic radii are from Shannon, 1976 [25]. # is from the single crystal high-pressure XRD [26]. * is from the present study. –: not provided in the reference.

| A- | Sm [24] | Eu [23] | $Gd_{1.5}Ce_{0.5}$* | Gd [11] | Tb [27] | Dy [23] | Y [27] | Ho [27] | Yb [28] |
|---|---|---|---|---|---|---|---|---|---|
| $r_A/r_B$ | 1.784 | 1.762 | 1.779 | 1.741 | 1.719 | 1.698 | 1.684 | 1.678 | 1.628 |
| Ps (GPa) | 40 | 42 | 39.8 | 38 | 39 | 40 | 42 | 37 | 28 |
| $B_0$ (GPa) | 185.4(2)# | — | 185(1) | 176(4) | 199(1) | — | 204(3) | 213(2) | 219(6) |
| $B_0'$ | 4.2 | — | 4(fixed) | 6.9(1.0) | 4(fixed) | — | 4.2(0.2) | 4(fixed) | 3.2(5) |

perspective, the pressure-induced phase transition from the pyrochlore phase to the cotunnite-like phase is an order–disorder transition. When $Gd^{3+}$ was replaced by $Ce^{3+}$, the average cationic radius of the A site and the ratio $r_A/r_B$ increased. Theoretical calculations have also proved that pyrochlore oxide with a larger $r_A/r_B$ causes higher defect formation energy (DFE) of cation antisite and anion Frenkel defects [29]. The higher DFE hinders the order–disorder transition. The results also confirm the substitution of large cationic radius actinides in the A-site of pyrochlore oxides will increase the cationic radius ratio $r_A/r_B$, and the transition pressure. But the substitution in the B-site will decrease the stability of the pyrochlore phase by lowering the $r_A/r_B$. So during the immobilization of the high level toxic nuclear waste, the A-site substitution will obtain a more stable form.

Silicate pyrochlore ($B^{4+} = Si^{4+}$) has been synthesized under high pressure and high temperature (for example, $Sc_2Si_2O_7$, $In_2Si_2O_7$ and $MgZrSi_2O_7$) [30,31]. These are composed of a larger A site cation and a much smaller B site cation ($Si^{4+}$), and their average cationic ratio is $r_A/r_B > 1.78$. Moreover, Si–O bonds in silicate pyrochlore are more likely to form covalent bonds, which are possibly hard to break. Accompanied by the above, these silicate pyrochlore oxides should transfer to the cotunnite-like structure at a much higher pressure.

Structure distortion is observed at approximately 9 GPa in many other titanite pyrochlores (e.g. $Gd_2Ti_2O_7$ [18] and $Tb_2Ti_2O_7$ [19]). At approximately 9 GPa, the 48$f$ $O^{2-}$ moves towards the vacancies and distorts the [$TiO_6$] octahedral, and this distortion is thought to be related to the pressure-induced crystallization of the spin liquid [32]. The compression behaviour of $Gd_{1.5}Ce_{0.5}Ti_2O_7$ is rather complex as a function of pressure. The pressure variation of the d-spacing for some strong diffraction peaks displayed twice change in slope: 6.5 GPa and 13.5 GPa, as shown in figure 6. To obtain the unit-cell parameters of $Gd_{1.5}Ce_{0.5}Ti_2O_7$ at various pressures in 2 runs (run 1: $p < 40$ GPa), the Le Bail refinement based on the pyrochlore structure for the *in situ* synchrotron X-ray diffraction patterns before the transition pressure was carried out and is plotted in figure 5, and the unit-cell volumes of various pressures are listed in table 2. There are three regions in the plot with distinctly different pressure dependencies. Due to the limited number of data in run 1, 2-nd Birch-Murnaghan EoS [33] was used

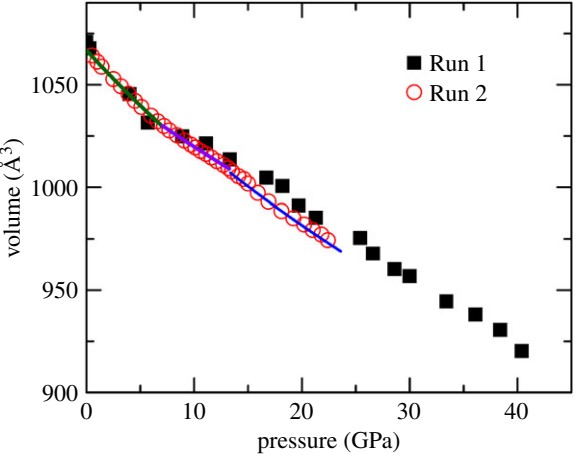

**Figure 5.** The $P$-$V$ relationship of $Gd_{1.5}Ce_{0.5}Ti_2O_7$. The black squares are the data of run 1, with silicone oil loaded as the pressure medium. The red circles are data from run 2, whose pressure medium is noble gas argon. The $P$-$V$ curves from three regions are shown in different colours: the dark green curve indicates $p < 6.5$ GPa, the purple curve shows $6.5$ GPa $< p < 13.5$ GPa and the blue curve is $p > 13.5$ GPa.

**Table 2.** The unit cell volumes and the $a$-axial lengths of the $Gd_{1.5}Ce_{0.5}Ti_2O_7$ under different pressures from run 1 and run 2.

| run 1 | | | run 2 | | | | | |
|---|---|---|---|---|---|---|---|---|
| $P$ (GPa) | $a$ (Å) | $V$ (Å$^3$) | $P$ (GPa) | $a$ (Å) | $V$ (Å$^3$) | $P$ (GPa) | $a$ (Å) | $V$ (Å$^3$) |
| 0.0001 | 10.231(1) | 1070.9(3) | 0.5 | 10.210(1) | 1064.3(2) | 12.1 | 10.042(1) | 1012.7(3) |
| 0.3 | 10.221(1) | 1067.8(3) | 1.0 | 10.200(1) | 1061.3(2) | 12.7 | 10.036(1) | 1010.9(4) |
| 4.0 | 10.149(1) | 1045.4(3) | 1.4 | 10.192(1) | 1058.8(2) | 13.2 | 10.031(1) | 1009.2(3) |
| 5.7 | 10.110(1) | 1031.5(4) | 2.5 | 10.173(1) | 1052.8(2) | 13.5 | 10.026(1) | 1007.7(3) |
| 8.9 | 10.082(1) | 1024.8(4) | 3.2 | 10.162(1) | 1049.3(2) | 14.1 | 10.018(1) | 1005.4(3) |
| 11.1 | 10.070(1) | 1021.3(3) | 3.9 | 10.150(1) | 1045.7(2) | 14.6 | 10.013(1) | 1004.0(3) |
| 13.3 | 10.045(1) | 1013.6(3) | 4.5 | 10.139(1) | 1042.2(2) | 15.0 | 10.006(1) | 1001.8(3) |
| 16.7 | 10.015(1) | 1004.6(3) | 5.1 | 10.129(1) | 1039.2(2) | 15.9 | 9.991(1) | 997.3(4) |
| 18.2 | 10.002(1) | 1000.6(4) | 6.0 | 10.115(1) | 1034.9(2) | 16.9 | 9.977(1) | 993.1(3) |
| 19.7 | 9.970(1) | 991.1(4) | 6.6 | 10.106(2) | 1032.2(2) | 18.1 | 9.962(2) | 988.5(3) |
| 21.3 | 9.951(1) | 985.3(3) | 7.2 | 10.099(1) | 1029.9(2) | 19.2 | 9.950(1) | 985.0(5) |
| 25.4 | 9.917(1) | 975.4(3) | 7.7 | 10.092(1) | 1027.7(2) | 20.2 | 9.939(1) | 982.0(5) |
| 26.6 | 9.892(2) | 967.8(5) | 8.4 | 10.084(1) | 1025.4(2) | 21.0 | 9.931(1) | 979.3(4) |
| 28.6 | 9.866(1) | 960.3(4) | 9.1 | 10.075(1) | 1022.7(2) | 21.8 | 9.923(2) | 977.1(4) |
| 30.0 | 9.853(1) | 956.8(3) | 9.7 | 10.069(1) | 1020.8(2) | 22.4 | 9.914(1) | 974.4(3) |
| 33.4 | 9.811(1) | 944.4(3) | 10.1 | 10.064(1) | 1019.4(2) | | | |
| 36.1 | 9.789(1) | 938.0(3) | 10.6 | 10.059(1) | 1017.6(2) | | | |
| 38.4 | 9.763(1) | 930.6(3) | 11.1 | 10.054(1) | 1016.2(2) | | | |
| 40.4 | 9.726(1) | 920.2(5) | 11.6 | 10.048(1) | 1014.5(3) | | | |

to fit these data only in run 2:

$$P = \frac{3}{2} B_0 \left[ \left(\frac{V_0}{V}\right)^{7/3} - \left(\frac{V_0}{V}\right)^{5/3} \right] \times \left\{ 1 + \frac{3}{4}(B'_0 - 4)\left[ \left(\frac{V_0}{V}\right)^{2/3} - 1 \right] \right\}, \tag{4.1}$$

where $p$ is the pressure, $B_0$ is the bulk modulus, $B'_0$ is the pressure derivative of $B_0$, $V_0$ is the unit-cell volume at zero pressure and room temperature. $B'_0$ is fitted to 4 for all of the data periods. The

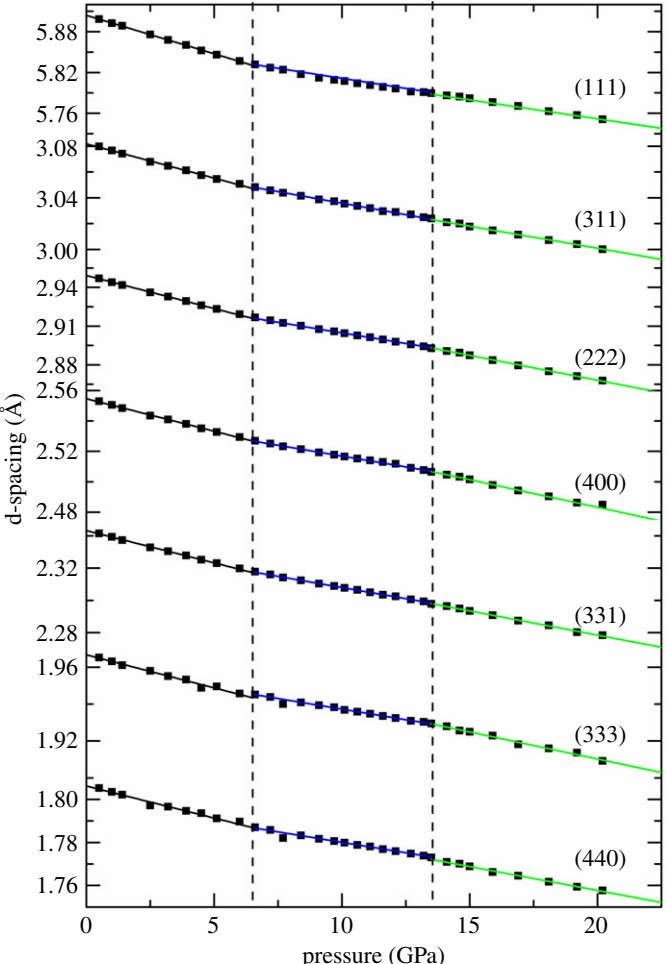

**Figure 6.** The d-spacing of $Gd_{1.5}Ce_{0.5}Ti_2O_7$ as a function of pressure. The black, blue and green lines are different regions ($p <$ 6.5 GPa, 6.5 GPa $< p <$ 13.5 GPa, and $p >$ 13.5 GPa, respectively).

**Table 3.** The parameters of the fitted EoS of the $Gd_{1.5}Ce_{0.5}Ti_2O_7$ ($B_0'$ is fixed to 4) and the $Gd_2Ti_2O_7$ from [18]. —: not provided in the reference.

| | $Gd_{1.5}Ce_{0.5}Ti_2O_7$ | | | $Gd_2Ti_2O_7$ | |
|---|---|---|---|---|---|
| $P$ | $0 \sim 6.5$ GPa | $6.5 \sim 13$ GPa | $>13$ GPa | $0 \sim 8.5$ GPa | $>8.5$ GPa |
| $B_0$ (GPa) | 185(1) | 261(2) | 195(5) | 176(4) | 208(8) |
| $B_0'$ | 4(fixed) | | | 6.9(1) | 1.0(3) |
| $V_0$ (Å$^3$) | 1066.9(1) | 1056.8(3) | 1070(2) | — | |

parameters of EoS are listed in table 3. Fitting the *P-V* curve before 6.5 GPa yields a bulk modulus of 185(1) GPa, which is compatible with pure $Sm_2Ti_2O_7$, but higher than $Gd_2Ti_2O_7$. Between 6.5 and approximately 13 GPa, the rate of change in the unit-cell volume is less than the region of $p < 6.5$ GPa, indicating an increase in the incompressibility. The $B_0$ of this region is 261(2) GPa, which increases by 40%. At pressures higher than 13 GPa, the slope again steepens. The bulk modulus over 13 GPa decreases to 195(5) GPa. In run 1, although the pressure transmitting medium is silicone oil, the same tendency is observed. Le Bail refinement of *in situ* high-pressure XRD (electronic supplementary material, figure S2) and *in situ* high-pressure Raman spectra confirm that no phase transition occurred. The d-spacing of each hkl of pyrochlore and the a-axial length *a* at various pressures also confirmed the compressibility changes at 6.5 GPa and 13 GPa. (figure 6; electronic supplementary material, figure S2). So we think there are two isostructural changes occurring in $Gd_{1.5}Ce_{0.5}Ti_2O_7$ at 6.5 GPa and 13 GPa.

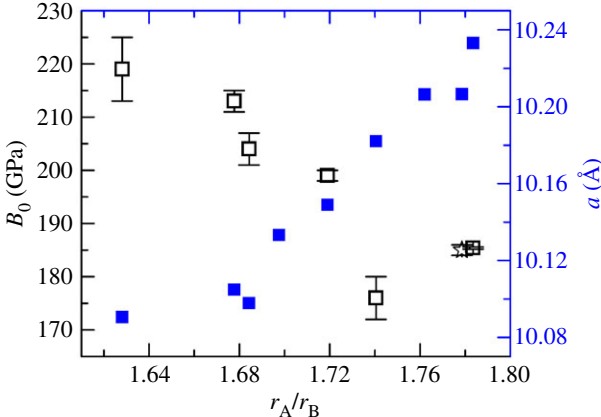

**Figure 7.** The bulk modulus of titanite pyrochlores and the unit-cell parameter *a* versus the cationic radii ratio ($r_A/r_B$). The black square marks represent the bulk modulus listed from previous research, the blue squares represent the *a* axial lengths, and the star mark represents the results of this study.

A similar phenomenon occurs in $Sm_2Zr_2O_7$ [34] and $La_2Zr_2O_7$ [35], and the mechanism is thought to be related to the anion disorder. Limited to the *in situ* high-pressure XRD experiment conditions, the refined crystal structure of the $Gd_{1.5}Ce_{0.5}Ti_2O_7$ could not be obtained. Besides, the high-pressure Raman spectrum is also not high quality enough to obtain the vibration frequencies at various pressures because the sample studied is polycrystalline powder, so a single crystal sample is essential. On the other hand, the electron structure of $Gd^{3+}$ is $[Xe]4f^7$, and $Ce^{3+}$ is $[Xe]4f^1$, which means the $Gd_{1.5}Ce_{0.5}Ti_2O_7$ is undoubtedly a strong correlation system. The possibility that this complex compression behaviour is caused by the transition of the f-electron structure is hard to rule out. Finally, compared with the *P-V* data plotted from run 1 and run 2, the compression behaviour changes of $Gd_{1.5}Ce_{0.5}Ti_2O_7$ may be related to the hydrostatic condition caused by the different PTM. Although the solidification pressure is 1.4 GPa at 300 K and its hydrostatic limit is approximately 9 GPa, Ar still provides a better hydrostatic condition than silicone oil does [36]. At P below 6.5 GPa, the unit-cell volumes of the samples are in good agreement in both run 1 and run 2. At P between 6.5 GPa and 13 GPa, the slope of the P-V curve from run 1 is lower than run 2, which means in silicone oil, $Gd_{1.5}Ce_{0.5}Ti_2O_7$ is more incompressible than in Ar. At *P* higher than 13 GPa, the P-V curve from run 1 is systematically higher than that from run 2, possibly due to the compressibility difference in the previous pressure regions. The slope of this region from the two runs is nearly the same. So the isostructural transitions may be related to the hydrostatic conditions. Above all, elucidating the mechanism of the complex compression behaviour of $Gd_{1.5}Ce_{0.5}Ti_2O_7$ requires more evidence.

Figure 7 shows the bulk modulus ($B_0$) when $P < P_c$, ($P_c$: the compressibility change pressure) of titanite pyrochlore oxides with different $r_A/r_B$. The bulk modulus $B_0$ of most of the titanite pyrochlores is higher than 180 GPa. The cationic radii ratio $r_A/r_B$ is negatively correlated to the bulk modulus. This is because the smaller A-site cationic radius can shorten the bond lengths by reducing the unit-cell parameters *a*. When they are shortening, the chemical bonds will be more incompressible. The average A-site cationic radius of $Gd_{1.5}Ce_{0.5}Ti_2O_7$ is similar to that of $Sm_2Ti_2O_7$ but larger than that of $Gd_2Ti_2O_7$. So the bulk modulus of $Gd_{1.5}Ce_{0.5}Ti_2O_7$ is quite similar to that of $Sm_2Ti_2O_7$ and higher than that of $Gd_2Ti_2O_7$.

# 5. Conclusion

The present experimental results demonstrate that ordered pyrochlore structured $Gd_{1.5}Ce_{0.5}Ti_2O_7$ ($Fd\bar{3}m$, $Z = 8$) will transfer to a disordered cotunnite-like structure ($Pnma$ $Z = 4$) at approximately 42 GPa. Compared with $Gd_2Ti_2O_7$, 25% $Gd^{3+}$ substituted by $Ce^{3+}$ increases the transition pressure because the pressure-induced pyrochlore to cotunnite-like phase transition was mainly controlled by the cation order–disorder transition. Furthermore, $Gd_{1.5}Ce_{0.5}Ti_2O_7$'s compression behaviour is rather complex as a function of pressure. Two isostructural transitions occur at 6.5 GPa and 13 GPa, which influences the compressibility of $Gd_{1.5}Ce_{0.5}Ti_2O_7$, and the transition at 6.5 GPa may be related to the hydrostatic conditions.

Supplementary materials. The supplementary materials associated with this article can be found in the online version: https://doi.org/10.6084/m9.figshare.9725534.v1.

Data accessibility. Our data are deposited at Dryad: https://doi.org/10.5061/dryad.14g7bk7 [37].

Authors' contributions. H.Z. and J.N. conceived of and designed the study. J.N. participated in experiments and drafted the manuscript. X.W. carried out XRD data analysis. S.Q. helped draft the manuscript and Raman spectra analysis.

Competing interests. The authors declare no competing interests.

Fundings. This work was financially supported by the National Natural Science Foundation of China (grants no. 91326102 and 51532009), and the Science and Technology Development Foundation of China Academy of Engineering Physics (grant no. 2013A0301012).

Acknowledgments. We thank Dr Hongrui Ding of Peking University for his assistance with the Raman experiment and providing comments that improved the manuscript. H.Z. is grateful to the foundation by the Recruitment Program of Global Youth Experts and the Youth Hundred Talents Project of Sichuan Province.

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
