## [Reviewer comments · Royal Society Open Science]

Review History

RSOS-190842.R0 (Original submission)

Review form: Reviewer 1

Is the manuscript scientifically sound in its present form?

Yes

Are the interpretations and conclusions justified by the results?

Yes

Is the language acceptable?

Yes

Do you have any ethical concerns with this paper?

No

Have you any concerns about statistical analyses in this paper?

No

Recommendation?

Accept with minor revision (please list in comments)

Comments to the Author(s)

In this work, the authors report an experimental study on ordered pyrochlore structured $\text{Gd}_{1.5}\text{Ce}_{0.5}\text{Ti}_2\text{O}_7$ by synchrotron radiation X-ray diffraction and Raman spectroscopy. An order-disorder transition is observed, and the transition is controlled by cationic order-disorder modification.

This manuscript could be considered for acceptance after a minor revision.

1. To better understand the phase transition behavior, especially the isostructural transitions at lower pressures, the authors could perform some theoretical calculations if possible.
2. The grammatical mistakes and other errors in the manuscript should be carefully revised.

Review form: Reviewer 2 (Chengming Fang)**Is the manuscript scientifically sound in its present form?**

Yes

Are the interpretations and conclusions justified by the results?

Yes

Is the language acceptable?

Yes

Do you have any ethical concerns with this paper?

No

Have you any concerns about statistical analyses in this paper?

No

Recommendation?

Accept as is

Comments to the Author(s)

In the current manuscript Niu and co-workers performed high-pressure experiments for $(\text{Gd}_{0.667}\text{Ce}_{0.333})_2\text{Ti}_2\text{O}_7$, a pyrochlore. They observed a phase transition at about 45GPa. They also reported two iso-structural transitions at 6.5 GPa and 13 GPa. The topic is interesting for people working in the related fields. The experimental details were described well. The results might be of interest for some people. The measured data were analyzed in a clear way. The manuscript was written concisely. The text is in the scope of the Journal. Therefore I'd like to propose acceptance of this manuscript for publication in Royal Society Open Science.

Decision letter (RSOS-190842.R0)

30-Jul-2019

Dear Dr Niu:

Title: Pressure-Induced Order-Disorder Transition in $\text{Gd}_{1.5}\text{Ce}_{0.5}\text{Ti}_2\text{O}_7$ Pyrochlore
Manuscript ID: RSOS-190842

Thank you for submitting the above manuscript to Royal Society Open Science. On behalf of the Editors and the Royal Society of Chemistry, I am pleased to inform you that your manuscript will be accepted for publication in Royal Society Open Science subject to minor revision in accordance with the referee suggestions. Please find the reviewers' comments at the end of this email.

The reviewers and handling editors have recommended publication, but also suggest some minor revisions to your manuscript. Therefore, I invite you to respond to the comments and revise your manuscript.

Because the schedule for publication is very tight, it is a condition of publication that you submit the revised version of your manuscript before 08-Aug-2019. Please note that the revision deadline will expire at 00.00am on this date. If you do not think you will be able to meet this date please let me know immediately.

Supplementary files will be published alongside the paper on the journal website and posted on

the online figshare repository (<https://figshare.com>). The heading and legend provided for each supplementary file during the submission process will be used to create the figshare page, so please ensure these are accurate and informative so that your files can be found in searches. Files on figshare will be made available approximately one week before the accompanying article so that the supplementary material can be attributed a unique DOI.

Best wishes,
Dr Laura Smith
Publishing Editor, Journals

RSC Associate Editor:
Comments to the Author:
(There are no comments.)

RSC Subject Editor:
Comments to the Author:
(There are no comments.)

Reviewer comments to Author:
Reviewer: 1

Comments to the Author(s)

In this work, the authors report an experimental study on ordered pyrochlore structured $\text{Gd}_{1.5}\text{Ce}_{0.5}\text{Ti}_2\text{O}_7$ by synchrotron radiation X-ray diffraction and Raman spectroscopy. An order-disorder transition is observed, and the transition is controlled by cationic order-disorder modification.

This manuscript could be considered for acceptance after a minor revision.

1. To better understand the phase transition behavior, especially the isostructural transitions at lower pressures, the authors could perform some theoretical calculations if possible.
2. The grammatical mistakes and other errors in the manuscript should be carefully revised.

Reviewer: 2

Comments to the Author(s)

In the current manuscript Niu and co-workers performed high-pressure experiments for $(\text{Gd}_{0.667}\text{Ce}_{0.333})_2\text{Ti}_2\text{O}_7$, a pyrochlore. They observed a phase transition at about 45GPa. They

also reported two iso-structural transitions at 6.5 GPa and 13 GPa. The topic is interesting for people working in the related fields. The experimental details were described well. The results might be of interest for some people. The measured data were analyzed in a clear way. The manuscript was written concisely. The text is in the scope of the Journal. Therefore I'd like to propose acceptance of this manuscript for publication in Royal Society Open Science.

Author's Response to Decision Letter for (RSOS-190842.R0)

See Appendix A.

Decision letter (RSOS-190842.R1)

12-Aug-2019

Dear Dr Niu:

Title: Pressure-Induced Order-Disorder Transition in $\text{Gd}_{1.5}\text{Ce}_{0.5}\text{Ti}_2\text{O}_7$ Pyrochlore
Manuscript ID: RSOS-190842.R1

It is a pleasure to accept your manuscript in its current form for publication in Royal Society Open Science. The chemistry content of Royal Society Open Science is published in collaboration with the Royal Society of Chemistry.

RSC Associate Editor

Comments to the Author:

The manuscript can now be accepted as is.

Reviewer(s)' Comments to Author:

Appendix A

Dear editor and reviewers,

Thank you for your comments concerning our manuscript entitled “Pressure-Induced Order-Disorder Transition in $\text{Gd}_{1.5}\text{Ce}_{0.5}\text{Ti}_2\text{O}_7$ Pyrochlore”. Following the comments, we have made corrections in the new revision. Finally, on behalf of all the authors, I would like to express my gratitude to the editors and the two reviewers.

The point-by-point response:

Referee: 1

1. To better understand the phase transition behavior, especially the isostructural transitions at lower pressures, the authors could perform some theoretical calculations if possible.

Theoretical calculations to reveal the high pressure behavior mechanism of the $\text{Gd}_{1.5}\text{Ce}_{0.5}\text{Ti}_2\text{O}_7$ pyrochlore is a challenging topic. However, we are sorry to say that we don't have enough resources to perform it. The reasons are following: First is the confusing magnetic structure. In $\text{Gd}_2\text{Ti}_2\text{O}_7$ pyrochlore, because of the topology of the crystal structure, the spins of the Gd^{3+} are geometrically frustrated. The magnetic structure of $\text{Gd}_2\text{Ti}_2\text{O}_7$ is still a currently active research topic. What's more, Ce^{3+} replacing Gd^{3+} will doubtfully effect the interaction of Gd^{3+} . So to determine the ground state magnetic structure of the $\text{Gd}_{1.5}\text{Ce}_{0.5}\text{Ti}_2\text{O}_7$ requires a lot of testing, and there is rarely experimental evidence can be consulted. Second, to perform the calculations on the $\text{Gd}_{1.5}\text{Ce}_{0.5}\text{Ti}_2\text{O}_7$ pyrochlore's high pressure behavior, we lack enough computing resources. The crystal structure of the high pressure stable cotunnite-like phase is a highly disordered structure. Both cations and anions are randomly distributed on 4c positions in the cotunnite-like structures, and 1/8 of the anions are missing. In previous studies (Xiao *et al*, 2009, Xiao *et al*, 2010), the structure of the cotunnite-like phase is established by the special quasirandom structure approach by using the Monte-Carl simulated. Unfortunately, we don't have enough source now. Besides, researchers believe that the similar isostructural transition in $\text{Sm}_2\text{Zr}_2\text{O}_7$ may be caused by the anion disorder (Zhang *et al*, 2007), and describing this anion disordered structure also requires a lot of computing resources. So it is hard for us to carry out theoretical calculations for better understand the high pressure behavior of the $\text{Gd}_{1.5}\text{Ce}_{0.5}\text{Ti}_2\text{O}_7$, but we have also found that it is an interesting and challenging topic worth carrying out. We will perform the theoretical research in the future.

References:

Xiao, H. Y., Fei Gao, and William J. Weber. "Ab initio investigation of phase stability of $Y_2Ti_2O_7$ and $Y_2Zr_2O_7$ under high pressure." *Physical Review B* 80.21 (2009): 212102.

Xiao, H.Y., et al. "Zirconate pyrochlores under high pressure." *Physical Chemistry Chemical Physics* 12.39 (2010): 12472-12477.

Zhang, F. X., et al. "Structural distortions and phase transformations in $Sm_2Zr_2O_7$ pyrochlore at high pressures." *Chemical physics letters* 441.4-6 (2007): 216-220.

2. The grammatical mistakes and other errors in the manuscript should be carefully revised.

They have been corrected.

P.S.

Jingjing Niu now works in *Institute of Tibetan Plateau Research, Chinese Academy of Sciences*. So his institution and e-mail address has been changed in the revised manuscript. This change has been agreed by the other co-authors.

We appreciate for Editors and Reviewers' warm work earnestly, and hope that the correction will meet with approval.

Once again, thank you very much for your comments and suggestions.

Jingjing Niu

Institute of Tibetan Plateau Research, Chinese Academy of Sciences.

Beijing, 100101, P.R.China.